# Stimulated Expression of CXCL12 in Adrenocortical Carcinoma by the PPARgamma Ligand Rosiglitazone Impairs Cancer Progression

**DOI:** 10.3390/jpm11111097

**Published:** 2021-10-27

**Authors:** Giulia Cantini, Laura Fei, Letizia Canu, Elena Lazzeri, Mariangela Sottili, Michela Francalanci, Maria Lucia Angelotti, Giuseppina De Filpo, Tonino Ercolino, Stefania Gelmini, Monica Mangoni, Gabriella Nesi, Constanze Hantel, Massimo Mannelli, Mario Maggi, Michaela Luconi

**Affiliations:** 1Department of Experimental and Clinical Biomedical Sciences “Mario Serio”, University of Florence, 50139 Florence, Italy; giulia.cantini@unifi.it (G.C.); laura.fei@unifi.it (L.F.); letizia.canu@unifi.it (L.C.); elena.lazzeri@unifi.it (E.L.); mariangela.sottili@unifi.it (M.S.); marialucia.angelotti@unifi.it (M.L.A.); giuseppina.defilpo@virgilio.it (G.D.F.); stefania.gelmini@unifi.it (S.G.); monica.mangoni@unifi.it (M.M.); massimo.mannelli@unifi.it (M.M.); mario.maggi@unifi.it (M.M.); 2Azienda Ospedaliero-Universitaria Careggi, 50139 Florence, Italy; tonino.ercolino@unifi.it; 3Meyer Children’s Hospital, 50139 Florence, Italy; michela.francalanci@libero.it; 4Department of Health Sciences, University of Florence, 50139 Florence, Italy; gabriella.nesi@unifi.it; 5Department of Endocrinology, Diabetology and Clinical Nutrition, University Hospital Zurich (USZ), University of Zurich (UZH), CH-8091 Zurich, Switzerland; Constanze.Hantel@usz.ch; 6Medizinische Klinik und Poliklinik III, University Hospital Carl Gustav Carus Dresden, 01307 Dresden, Germany; 7I.N.B.B. (Istituto Nazionale Biostrutture e Biosistemi), 00136 Rome, Italy

**Keywords:** chemokines, rare cancers, ACC, xenograft cancer models, anti-cancer therapy, thiazolidinediones

## Abstract

Adrenocortical carcinoma (ACC) is a rare malignancy with poor prognosis when metastatic and scarce treatment options in the advanced stages. In solid tumors, the chemokine CXCL12/CXCR4 axis is involved in the metastatic process. We demonstrated that the human adrenocortex expressed CXCL12 and its cognate receptors CXCR4 and CXCR7, not only in physiological conditions, but also in ACC, where the receptors’ expression was higher and the CXCL12 expression was lower than in the physiological conditions. In a small pilot cohort of 22 ACC patients, CXCL12 negatively correlated with tumor size, stage, Weiss score, necrosis, and mitotic activity. In a Kaplan–Meier analysis, the CXCL12 tumor expression significantly predicted disease-free, progression-free, and overall survival. In vitro treatment of the primary ACC H295R and of the metastatic MUC-1 cell line with the PPARγ-ligand rosiglitazone (RGZ) dose-dependently reduced proliferation, resulting in a significant increase in CXCL12 and a decrease in its receptors in the H295R cells only, with no effect on the MUC-1 levels. In ACC mouse xenografts, tumor growth was inhibited by the RGZ treatment before tumor development (prevention-setting) and once the tumor had grown (therapeutic-setting), similarly to mitotane (MTT). This inhibition was associated with a significant suppression of the tumor CXCR4/CXCR7 and the stimulation of human CXCL12 expression. Tumor growth correlated inversely with CXCL12 and positively with CXCR4 expression, suggesting that local CXCL12 may impair the primary tumor cell response to the ligand gradient that may contribute to driving the tumor progression. These findings indicate that CXCL12/CXCR4 may constitute a potential target for anti-cancer agents such as rosiglitazone in the treatment of ACC.

## 1. Introduction

Adrenocortical carcinoma (ACC) is a rare and aggressive malignancy of the steroidogenic component of the adrenal gland. To date, tumor radical resection is the only curative option. Treatment with the adrenolytic drug mitotane (MTT) has been shown to increase overall survival (OS), particularly in advanced stages [1,2]. The risk of short-term recurrence is high, leading to a mortality rate of up to 80% in metastatic ACC at 5 years from surgery [3]. Specific and effective anti-cancer therapies are therefore urgently needed to target the metastatic progression of this tumor.

The chemokine axis between the C-X-C motif chemokine 12 (CXCL12) ligand and its two receptors, C-X-C-Motif-Chemokine-Receptor (CXCR)-4 (CXCR4) and -7 (CXCR7), is actively involved in controlling the metastatic process of several solid tumors [4]. High levels of CXCL12 are expressed at common sites of cancer metastasis, such as lung, liver, and bone, and the occurrence of the gradients of this chemokine prompts local invasion and subsequent homing from the primary tumor [5] to the secondary sites of metastasis [4].

CXCR4 and CXCR7 expression has also been described in ACC [6], and high levels of the two receptors have been detected in ACC metastases [7,8]. However, semi-quantitative immunohistochemical evaluation of the receptors in the primary tumor biopsies found no significant correlation with patient clinical parameters or any prognostic value [6]. Despite the fact that CXCL12 can promote the detachment of cancer cells from the primary tumor and their entry into the bloodstream in response to a metastatic site-driven gradient, the role of this cytokine in controlling tumor progression and metastasis in primary ACC has not yet been assessed.

Interestingly, the glucose-lowering drug rosiglitazone (RGZ), a thiazolidinedione ligand of the peroxisome proliferator-activated receptor gamma (PPARγ), previously shown to affect ACC tumor growth and invasiveness in preclinical studies [9,10,11,12], has more recently proven to exert its anti-cancer activity by interfering with the CXCL12/CXCR4 axis in colon and prostate cancer cells [13,14].

The aim of this study was to elucidate the role of the CXCL12 ligand in ACC as a potential target for RGZ. The experimental approach combined in vitro experiments on two different primary and metastatic cell line models of ACC, with the analysis of cancer biopsy samples from a small pilot cohort of ACC patients and ACC mouse xenograft models in order to assess RGZ in vivo activity.

## 2. Results 

### 2.1. CXCL12 and Its Cognate Receptors Are Differentially Expressed in ACC and Normal Adrenocortex

The differential quantitative expression of the CXCR4/CXCR7 receptors and their ligand CXCL12 was evident between the tumor samples from a pilot cohort of n = 22 ACC patients and n = 26 normal adrenals. To investigate any possible relation between paracrine tumor activity and the expression of the CXCL12/CXCR4 axis, IGF-II was also quantified. The anthropometric and main clinical characteristics of the ACC patients are detailed in Table 1. 

The ACC expressed one-log-unit higher levels of CXCR4 and CXCR7 compared to the normal adrenals, whereas CXCL12 was expressed at higher levels in the normal adrenals than in the ACCs (Figure 1A). An increased IGF-II expression was typical of the ACCs (Figure 1A). The CXCR4 differential expression between the ACC and the normal adrenals (FI ACC vs. NOR: 32.5 ± 16.7-fold, *p* < 0.0001) was also established at protein level by a Western blot analysis of the protein lysates (Figure 1B). A CXCR4 protein band was also present in the lysates from the H295R cell line (Figure 1B). 

Immunofluorescence confocal analysis performed on the ACCs and the normal adrenals confirmed the differential expression of the receptors and ligand between the neoplastic and the normal tissues (Figure 2, left panel). Positivity for CXCR4 (Figure 2A,B) and CXCR7 (Figure 2C,D) was predominantly localized at the level of the adrenal parenchyma and showed a pronounced intensity in the ACCs (Figure 2B,D) compared to the normal adrenals (Figure 2A,C). CXCL12 was expressed at a higher level in the normal adrenals (Figure 2E) compared to the ACC samples (Figure 2F), not only in the vasculature (markedly positive in normal adrenals, Figure 2E), but also in the normal (Figure 2E) and malignant adrenal (Figure 2F) cells. Immunofluorescence analysis of H295R cells revealed a membrane positivity for both CXCR4 and CXCR7 and a strong membrane and cytosol positivity for CXCL12 (Figure 2, right panel). 

### 2.2. Prognostic Value of CXCL12 Expression in ACC

To gain insight into the possible role of the CXCL12/CXCR4/CXCR7 axis in tumor development and progression, we explored any association between the expression of the two receptors, or of their ligand, and the clinical characteristics of ACC patients (see Table 1). No significant correlations were found between CXCR4 or CXCR7 and any of the ACC parameters. Conversely, a significant negative correlation was found between CXCL12 expression and tumor size (r = −0.504, *p* = 0.033), the stage (r = −0.528, *p* = 0.014), and the global Weiss score (r = −0.666, *p* = 0.001), but not with Ki67, as well as two of the Weiss parameters, necrosis (r = −0.906, *p* < 0.001), and the mitotic figures (r = −0.693, *p* = 0.004). By stratifying the ACC-patient population according to the upper tertile value of CXCL12 mRNA distribution in low (≤169, 2^−ΔΔCt^) and high (>169, 2^−ΔΔCt^) expression, Kaplan–Meier survival analysis showed that CXCL12 significantly predict DFS (log rank = 0.022, Figure 3A), PFS (log rank = 0.010, Figure 3B), and OS (log rank = 0.048, Figure 3C). 

In low- and high-CXCL12 groups, recurrence/metastasis, tumor progression and death events were 12 and 1, 10 and 1, and 9 and 0, respectively. Disease-free time and progression-free time were significantly increased in the high- vs. the low-CXCL12 group (81.9 ± 24.0 vs. 24.1 ± 8.3 months, *p* < 0.010 and 81.9 ± 24.0 vs. 24.6 ± 8.3 months, *p* < 0.010, respectively), while the increase in OS did not reach any statistical significance (91.3 ± 23.2 and 59.3 ± 13.1 months, *p* = 0.211). Significant differences between the two groups were also found for tumor size (6.3 ± 1.5 vs. 11.0 ± 1.1, *p* = 0.035), Weiss score (4.3 ± 0.6 vs. 6.7 ± 0.4, *p* = 0.002), and age (38.3 ± 4.6 vs. 53.1 ± 3.8, *p* = 0.041), with the CXCL12 low-expressing group showing worse parameters (Table 1). High levels of CXCL12 tumor expression associated with a significantly reduced risk of recurrence/metastasis, with a Hazard Ratio [95% confidence interval] of 0.12 [0.01–0.91], *p* = 0.041, which remained statistically significant after adjustment for Ki67 0.07 [0.01–0.62], *p* = 0.017, the global Weiss score 0.09 [0.01–0.92], *p* = 0.043, and age 0.13 [0.01–1.07], *p* = 0.05, as confounding factors. Similar results were obtained for the risk of tumor progression (not shown). Conversely, the HR of death did not reach any statistical significance (not shown).

### 2.3. Rosiglitazone Upregulates CXCL12 Expression and Downregulates CXCR4/CXCR7 Expression in Adrenocortical Cancer Cells In Vitro and in In Vivo ACC Xenograft Mouse Models

As the PPARγ ligand RGZ has been shown to interfere with this axis in other solid tumors [13,14,15,16], we evaluated whether the CXCL12/CXCR4 axis might contribute in mediating the ACC anti-cancer activity of RGZ [9,11,12,17] in different ACC cell lines. In the primary tumor cell line model, H295R, as well as in the metastatic cell line, MUC-1, RGZ (20 and 50 µM) dose-dependently reduced the cell number (Figure 4A,B), whereas the effect was time-dependent only in the H295R cells. RGZ treatment was associated with a significant increase in CXCL12 and a decrease in CXCR4 and CXCR7 expression only in the primary tumor H295R cell line (Figure 4C), while no significant effects were observed in the metastatic MUC-1 cell line (Figure 4D). Notably, significant differences were detected in the baseline expression levels of both the receptors and the CXCL12 ligands between H295R and MUC-1 cells (MUC-1 vs. H295R FI: CXCR4 44.7 ± 5.6-fold, *p* = 0.0001; CXCR7 10^−4^ ± 10^−5^ -fold, *p* = 0.0001; and CXCL12 2.4 ± 0.7-fold, *p* = 0.01).

We then assessed the effect of RGZ in vivo in ACC xenograft models obtained by the subcutaneous injection of H295R cells in immunocompromised mice. We compared the effects of RGZ on the tumor mass growth with those attained with MTT, the current chemotherapeutic agent used in ACC, alone and in combination with RGZ. Three different models of mouse xenograft were set up, as described in Figure 5. First, a “therapeutic model”, where RGZ or MTT, or their combination, were given to the animals only when the H295R-induced tumor had developed as a subcutaneous mass of about 4 mm in diameter (Figure 5, upper panel). We also developed a “prevention model”, where immunocompromised mice were pre-treated for 7 days with RGZ or MTT before being injected with H295R cells, and treatment was continued until animal sacrifice (animal pre-treatment model, Figure 5, middle panel). A second “prevention model” was acquired by the subcutaneous injection of H295R cells pre-treated in vitro for 24 h with 20 µM RGZ, and no subsequent RGZ-animal treatment (cell pre-treatment) (Figure 5, lower panel).

Analysis of the tumor growth curves in the therapeutic model showed that RGZ and MTT alone, or combined, were able to significantly reduce tumor growth compared to the untreated animals, starting from day 8 for all treatments (Figure 6A). No significant difference between the various treatment arms was observed (Figure 6A). At day 26, tumor growth inhibition was 64%, 58%, and 50% for RGZ, MTT, and RGZ + MTT, respectively (Figure 6A). In the animal pre-treatment xenograft model, where RGZ and MTT treatments started one week before H295R cell injection and continued with the same protocol until animal sacrifice (Figure 6B), both treatments similarly inhibited tumor growth compared to the untreated control animals. At day 48, the inhibitory effect was 61% and 75% for RGZ and MTT, respectively. Finally, to evaluate a possible direct effect of RGZ on the H295R cell proliferative and invasive mechanisms, the H295R cells were pre-treated for 24 h with 20 µM RGZ before subcutaneous injection, with no further animal treatment. We compared the effects of cell- and animal-pre-treatment (Figure 6C). We found that the inhibitory effect on the mass growth exerted by the RGZ-animal pre-treatment vs. the untreated controls was significant at any time point and higher than the corresponding effects achieved by RGZ-cell treatment before the injection (Figure 6C). 

In the therapeutic model, upon animal sacrifice, the tumor masses were excised and subjected to qRT-PCR analysis of the human and mouse genes in order to assess the effects of the RGZ and MTT in vivo treatment on the CXCL12/CXCR4/CXCR7 axis and tumor vascularization.

The RGZ and MTT similarly inhibited human CXCR4 and, to a lesser extent, human CXCR7 expression with no additive effect when the treatments were combined (Figure 7), compared with the untreated animals. This was accompanied by a significant increase in CXCL12 expression, more evident for MTT, again with no additive effect when RGZ and MTT were administered together (Figure 7). A significant negative linear correlation was found between tumor growth and CXCL12 (r = −0.573, r2 = 0.329, *p* = 0.008, n = 20), while a positive linear correlation emerged with CXCR4 (r = 0.448, *p* = 0.037, r2 = 0.201, n = 23). No significant association was found with CXCR7. The two drugs not only affected the human component of the tumor mass but also its vascularization, given that the VEGF and CD31 expression was also markedly reduced in the treated animals (Figure 7).

## 3. Discussion

This is the first study evaluating the expression and role of CXCL12 and its cognate receptors in ACC, along with their association with the anti-cancer activity of RGZ.

We demonstrated a differential expression of the axis components between malignancy and normal adrenal, showing a reduced expression of CXCL12 protein in the tumor compared with the normal adrenocortex, and confirming the CXCR4 and CXCR7 expression as reported in the literature [6]. In our paper, we also demonstrated that both the primary tumor H295R cell line and the metastatic MUC-1 cells produce CXCL12 and express its receptors. These data indicate that both the normal adrenocortical parenchyma and the tumor cells actively produce CXCL12, which can perform locally in a paracrine manner in pathophysiological conditions. RGZ is able to up-regulate CXC12 only in primary, but not in metastatic, cells.

Although we found no correlation between the receptor expression and the ACC clinical parameters, the CXCL12 mRNA levels were inversely correlated with tumor size, stage, Weiss score, tumor necrosis, and mitotic activity. The Kaplan–Meier analysis of survival in patients stratified into high- and low-expressing tumors, revealed that CXCL12, but not CXCR4 and CXCR7, could significantly predict DFS and OS. Cox regression univariate and multivariate analysis adjusted for Ki67 and Weiss score further suggested that high CXCL12 local expression is associated with less aggressive tumors and confers protection against tumor progression and recurrence. In agreement with our data, Chifu and colleagues recently found no association of CXCR4 and CXCR7 with metastasis or OS in a large series of ACCs, although these findings were semi-quantitative, as the authors only evaluated the immunohistochemical positivity of the two receptors [6]. Our findings indicate that in ACC the tumor expression of the ligand rather than that of its receptors is more strongly associated with progression and aggressiveness. The fact that intratumor CXCL12 production actively interferes with cancer progression is further supported by previous findings in breast and bone cancer. In fact, in breast cancer patients lower CXCL12 expression levels correlated with worse prognosis, mirrored by higher levels of the ligand detected in breast cancer cell lines with a lower metastatic potential [18,19]. Similarly, high CXCL12 immunohistochemical expression resulted in a better OS outcome and prognosis in osteosarcoma [20]. The causative effects of intratumor CXCL12 are suggested by the epigenetic silencing of the CXCL12 promoter, which resulted in high metastatic potential in breast cancer cells [21], while the re-expression of CXCL12 in highly aggressive mammary carcinoma cell lines limited metastatic progression in mouse xenograft models [21].

Interestingly, an in vitro co-culturing model of the adipose stem precursors and H295R cells, mimicking the adipose tissue infiltration by the tumor cells observed in advanced ACC, enhanced the invasive properties of the H295R cells and was accompanied by a decrease in the CXCL12 expression in the tumor cells [22]. These findings indicate that the stimulation of cancer cell invasiveness may involve a decrease in endogenous CXCL12 production to respond to the ligand gradient established by the adipose-tissue microenvironment [22]. 

In addition to the mechanism by which the elevated local levels of CXCL12 interfere with the cancer cell migration in response to a chemokine gradient towards the site of metastasis, a CXCL12 effect on T cell immune response has also been advocated in osteosarcoma [20]. The CXCL12 concentration was associated with an increase in the number of intra-tumor lymphocytes in the osteosarcoma samples, highlighting the potential role of CXCL12 in lymphocyte homing to the primary tumor [20]. Innate immune response has also been suggested in mediating the CXCL12 production in a Ptenfl/fl Trp53fl/fl mouse model of prostate cancer after cabozantinib treatment. Cabozantinib triggers the release of CXCL12, resulting in robust neutrophil infiltration into the tumor bed and near-complete clearance of poorly differentiated murine prostate cancer [23]. This additional effect of CXCL12 on infiltrating immune cells also deserves to be assessed in ACC.

The adrenal is often considered an immune-privileged organ, due to the local production of corticosteroids. Patients with cortisol-producing ACC have a worse prognosis [24], and glucocorticoid excess is associated with impaired immune infiltration and T cell depletion, which may contribute to tumor progression [25]. Further studies are mandatory to assess whether hypercortisolism is somehow associated with a reduction of CXCL12 levels and a decreased local immune response in ACC.

In order to modulate the CXCL12/CXCR4 system in ACC, we evaluated the effects on the axis exerted by the PPARγ ligand RGZ, an anti-diabetic drug we previously demonstrated able to inhibit ACC growth in vitro and in vivo by interfering with the PI3K and ERK activity [11,12,17]. In breast cancer cells, RGZ inhibits cell invasiveness by blocking the CXCR4 downstream signaling involving FAK, Akt, and ERK [16]. This anti-cancer effect seems to be mediated by the activity of the ligand-activated PPARγ complex, as a novel PPAR responsive element has been predicted on the CXCR4 promoter [16]. Here, we showed that RGZ dose-dependently inhibited the in vitro proliferation of the adrenocortical carcinoma primary tumor cell line, H295R, as well as of the metastatic cell line, MUC-1. The effect was similar between the two cell lines at two days, while the MUC-1 seemed to be more resistant than the H295R cells in the longer period. A different sensitivity in the response to cytotoxic agents has already been reported for MUC-1 and H295R cells, with MUC-1 showing to be more chemoresistant than the primary tumor cells due to their metastatic origin [26,27,28,29,30]. The reduced proliferation was paralleled by a significant increase in the CXCL12 gene expression and the decrease in its two receptors only in H295R cells. Conversely, in MUC-1, RGZ did not modulate the CXCL12/CXCR4 axis, suggesting that the up-regulated CXCL12 in response to RGZ might be active only in the primary tumor but not in the metastatic cells. These findings, together with the high levels of CXCL12 in the tumor associated with a lower risk of metastasis and progression, suggest that in the primary tumor, CXCL12 endogenously produced by the tumor cells might act in a paracrine manner to contain tumor progression, making tumor cells less sensitive to the CXCL12 gradient that drives the metastatic process. The baseline CXCL12 production in the metastatic cells might contribute to the generation of the chemokine gradient that attracts cancer cells to the site of the metastasis. Cells from the primary tumor and metastasis display differences in the levels of the cognate receptors, as the H295R cells express both receptors, while the MUC-1 ones only CXCR4 at a significantly higher level than in the other cell model. Both receptors are important for cell migration, however CXCR7 with its higher affinity for CXCL12 seems to mediate interaction with the endothelial cells, facilitating transendothelial migration from the primary mass [31], whereas CXCR4 may be more relevant for driving cells along the ligand gradient for the metastasis. Moreover, the ability of CXCR7 to modulate the CXCR4 response to CXCL12 through competition on the downstream signaling pathways involving β-arrestin 2 [32] may be relevant for the intratumor paracrine activity of CXCL12.

We developed different ACC xenograft mouse models to study the RGZ effects in vivo on tumor growth and the CXCL12/CXCR4 axis in a therapeutic and a prevention setting, comparing them with MTT. In the therapeutic model, a significant inhibition of the tumor growth was achieved early and to a similar extent with RGZ and MTT. The absence of additive effects when the two drugs were used together suggests that RGZ and MTT may converge on the same signaling pathways controlling cell proliferation, though their identification deserves further studies. Since CXCL12 production is stimulated by estradiol in ovarian and breast cancer [33] and mitotane’s effects have been described to be partially mediated by the activation of the estradiol receptors [34] also in ACC [27], mitotane-induced CXCL12 production in ACC might at least partially involve estrogen-like activity. Tumor-growth inhibition was associated with the increased expression of CXCL12 and the decreased expression of the two receptors in the human component of the tumor mass, along with a reduced vascularization of the tumor. The strong negative correlation found between tumor growth and CXCL12 expression in cancer cells suggests a major role of CXCL12 in controlling tumor progression. RGZ and MTT also prevented tumor growth when animals were treated before H295R cell injection, implying an interference of this treatment with cell engraftment and growth. The suppression of tumor growth in the ACC xenograft model obtained by subcutaneous injection of RGZ-pre-treated H295R cells suggests that the inhibitory activity of RGZ may be mediated by a direct effect on cancer cells.

The limitations of our study are the small number of patients analyzed and the lack of experiments performed with specific inhibitors of the CXCL12/CXCR4 axis to support fully causative effects. Although CXCR4 antagonists have already been tested in clinical trials in other type of cancers, demonstrating that blocking CXCL12 binding to CXCR4 can interfere with cancer progression [35,36], specific blockers of CXCL12 production are still lacking. Instead, local use of dipeptidyl peptidase-4 (DPP4) inhibitors to block CXCL12-inactivating enzymes may be hypothesized to locally increase the ligand levels and further validate this role of this chemokine in impairing tumor progression toward metastasis.

## 4. Materials and Methods

### 4.1. Patients

The study comprises 22 ACC patients, whose clinical characteristics are detailed in Table 1. All patients underwent surgical removal of the tumor mass at Careggi University Hospital, Florence, Italy. Normal adrenal specimens were obtained during radical nephrectomy for renal carcinoma or from organ donors (n = 26). Informed consent was obtained from all subjects involved in the study. The study was approved by the Local Ethical Committee (Prot.2017-277 BIO 59/11, 27/09/2017) and follows the Code of Ethics of the World Medical Association (Declaration of Helsinki). The tumor (ACC) specimens and normal adrenal (NOR) samples were snap frozen and stored at −80 °C until protein and mRNA extraction or immunofluorescence analysis or were formalin-fixed and paraffin-embedded for immunohistochemistry.

### 4.2. Antibodies

The following antibodies were used for the axis identification [31]: anti-CXCR4 (clone 12G5, mouse IgG2a; R&D Systems); anti-CXCR7 (rabbit polyclonal IgG Affinity Bioreagents, ThermoFisher, MA, USA); anti-CXCL12 (clone 179018, mouse IgG1; R&D Systems, Minneapolis, MN, USA); anti-CD31 (clone WM59, mouse IgG1, Becton Dickinson Italia S.p.A., Milan, Italy); and anti-vWf (polyclonal rabbit IgG, Dako, Glostrup, Denmark). Alexa-Fluor secondary antibodies as 546-labeled goat anti-mouse IgG1; Alexa Fluor 488-labeled goat anti-mouse IgG2a; and Alexa Fluor 488-labeled goat anti-rabbit IgG were obtained from Molecular Probes, Life Technologies, Monza, Italy. Actin (goat polyclonal-C11 #sc1615) was from Santa Cruz Biotechnologies Inc. (Santa Cruz, CA, USA), whereas the anti-mouse and anti-goat peroxidase-secondary IgG used in the Western blot analysis were from Sigma-Aldrich (St. Louis, MO, USA). Anti-human Ki67 monoclonal MIB1 antibody was used for Ki67 Labeling Index (Ki67 LI) (Dako, Carpinteria, CA, USA).

### 4.3. Histologic Diagnosis and Immunohistochemistry

The histologic diagnosis of ACC was performed by the referent pathologist (G.N.) on the tumor tissue removed at the surgery. The tumor specimens were evaluated according to the Weiss System, where the presence of three or more criteria correlates highly with malignant behavior [37].

The Ki67 proliferation index was assessed using the anti-human Ki67 monoclonal MIB1 antibody (Dako, Carpenteria, CA, USA). Ki67 positive nuclei were counted on 1000 tumor cells and Ki67 LI was expressed as the percentage of proliferating cells.

Tumors were staged according to the revised TNM classification of ACC proposed by the European Network for the Study of Adrenal Tumors (ENS@T) [38].

### 4.4. Cell Culture and Growth Analysis

The human ACC cell lines H295R, obtained from the American Type Culture Collection (Manassas, VA, USA), were cultured in DMEM/F-12 medium (Sigma-Aldrich) with 10% FBS, 2 mM L-glutamine, and 100 U/mL penicillin-100 μg/mL streptomycin, enriched with a mixture of insulin/transferrin/selenium (Sigma-Aldrich). The MUC-1 cells, kindly provided by Dr Hantel, were cultured in Advanced DMEM/F12 medium (Thermo-Fisher) with 10% FBS, 2 mM L-glutamine, and 100 U/mL penicillin-100 μg/mL streptomycin as previously described [26]. The cells were incubated at 37 °C in a humidified 5% CO_2_ atmosphere.

H295R and MUC-1 cells seeded in 12-well plates (1 × 10^5^ cells/well) were 24 h-starved and treated in 10% FBS-medium with vehicle (control) or with RGZ for the indicated time points. At each time point, the cells were trypsinized and counted with a hemocytometer, after dead cell exclusion with a trypan blue test. The mean cell number was obtained by counting four replicates in three independent experiments.

### 4.5. RNA Isolation and Quantitative Real-Time PCR

The mRNA was isolated from frozen human ACC and normal adrenal tissues, excised xenografts, and harvested cells using the RNeasy Mini Kit (Qiagen, Hilden, Germany), as previously described [39]. 

For each RNA sample, cDNA was obtained by reverse transcription PCR starting from 250 ng of RNA in a 50 μL final volume reaction (Taqman RT-PCR kit; Applied Biosystems, Foster City, CA, USA) using the following cycling conditions: 10 min at 25 °C, 30 min at 48 °C, and 3 min at 95 °C, held at 4 °C. Further quantitative real-time PCR (qRT-PCR) was carried out using primers and probes from Applied Biosystems for the following gene transcripts: human CXCR4 (Hs00237052_m1); human CXCR7 (Hs00604567_m1); human CXCL12 (Hs00171022_m1); human IGF-II (Hs04188276_m1); GAPDH (4352934) and mouse CD31 (Mm00476712_m1); VEGF (Mm00437306_m1); and RSP18 (Mm02601777_g1). The RT-PCR reactions, performed in triplicate for each gene, were carried out on an ABI Prism 7900 Sequence Detector (Applied Biosystems). The amount of target genes, normalized to the endogenous reference gene (human GAPDH for human gene expression of H295R and MUC-1 or to mouse RSP18 for mouse gene expression in the excised tumors) and relative to a calibrator (Stratagene, San Diego, CA, USA), was calculated by 2^−ΔΔCt^.

### 4.6. Sodium Dodecyl Sulphate PolyAcrylamide Gel Electrophoresis (SDS-PAGE) and Western Blot Analysis

The treated cells were lysed in RIPA buffer (20 mM Tris pH 7.4, 150 mM NaCl, 0.2 mM EDTA, 0.5% Triton-100, 1 mM Na3VO4) supplemented with 100× phosphatase inhibitor and 100× protease inhibitor (Sigma-Aldrich). The tissue samples were homogenized by mechanical disruption with Ultraturrax T10 basic IKA (Werke GmbH & Co., Staufen, Germany) in RIPA lysis buffer. After protein measurement by the Coomassie method, equal amounts of proteins for each sample (30 µg) were separated by reducing SDS–PAGE and transferred onto PVDF membranes (BIO-RAD Labs, CA, USA).

The membranes were incubated overnight at 4 °C with anti-human CXCR4 and actin, followed by peroxidase-secondary IgG at room temperature. Image acquisition and densitometric analysis were performed with Quantity One software on a ChemiDoc XRS instrument (BIO-RAD Labs, CA, USA). All Western blots were repeated in at least three independent experiments. Actin was used as an internal loading control to normalize protein expression in each lane.

### 4.7. Immunofluorescence Confocal Microscopy Analysis of Tissues and Cells

Confocal microscopy was performed on 5 μm sections of normal adrenal and ACC frozen biopsies or on H295R cells cultured on chamber slides by using an LSM 510 META laser confocal microscope (Carl Zeiss, Inc.), as previously described (29). Immunofluorescence was performed with anti-CXCR4 (40 μg/mL dilution) or anti-CXCL12 (20 μg/mL dilution) antibodies in combination with anti-vWf antibodies (1.3 μg/mL dilution) or with anti-CXCR7 (10 μg/mL dilution) in combination with anti-CD31 antibodies (10 μg/mL), respectively. Appropriate Alexa-Fluor secondary antibodies were used (Molecular Probes, Life Technologies). TO-PRO-3 was used for nuclei counterstaining (Invitrogen, Carlsbad, CA, USA). All random scans of the adrenal tissue were recorded at the same photomultiplier tube, pinhole aperture, and laser voltage setting by using an LSM 510 META laser confocal microscope.

### 4.8. ACC Mouse Xenograft Models

Female athymic CD1 nude mice (9-week-old, Charles River Laboratories, Italy) were inoculated subcutaneously with H295R cell suspension (7.5 × 10^6^ cells/100 µL) to generate the ACC mouse models. The scheme of therapeutic and prevention xenograft models is illustrated in Figure 5. Tumor volume (mm^3^) was monitored 3 times a week by two independent investigators and was calculated by using the following formula: length × width^2^/2. Six animals/group were injected. Animal sex was not expected to affect the results.

Drug tolerability in tumor-bearing mice was assessed in terms of: (a) lethal toxicity, i.e., any death in the treated mice occurring before any death in the control mice; (b) body-weight loss percentage = 100—(body weight on day x/body weight on day 1) × 100, where x represents a day after or during the treatment period [12,40].

The animal studies were performed in compliance with an institutionally approved protocol and with the National Institutes of Health Guide for the Care and Use of Laboratory Animals (NIH Publications No. 8023, revised 1978). The animals were sacrificed at the indicated time and the tumor explant excised for qRT-PCR analysis.

### 4.9. Statistical Analysis

The data are reported as mean ± SE. The normal distribution of parameters was evaluated by Kolmogorov–Smirnov’s test. The comparison between the two groups of data was accomplished using the Student’s *t* test for continuous variables, whereas the χ^2^ test was applied for discrete variables. A U-Mann–Whitney test was performed for nonparametric variables. Multiple-group comparisons were carried out using ANOVA followed by Dunnett’s post-hoc test. Univariate correlation between parameters was conducted with Pearson’s test. Statistical significance for two-tailed analyses (*p*-value) was assigned for values < 0.05. OS, DFS, and PFS was defined as the probability (ranging from 0 to 1) that a patient diagnosed with the disease is still alive (OS), is free from the disease (DFS), or is free from progression (PFS) at a time point from surgery. Survival analysis was estimated through the Kaplan–Meier method, and the differences between the groups were assessed by log-rank test. Cox multiple regression analysis was used to study the factors considered to independently influence DFS. All statistical analyses were performed using SPSS Statistics Version 27 (IBM). 

## 5. Conclusions

Our findings open new perspectives for repurposing RGZ and other thiazolidinedione PPARγ ligands in the treatment of ACC to specifically target the CXCL12/CXCR4 axis in order to counteract cancer progression. The validation of these preliminary results in larger cohorts of ACC patients is mandatory, and further experiments must be conducted to better clarify the role of CXCL12 in mediating the anti-cancer effects of RGZ in ACC.

## Figures and Tables

**Figure 1 jpm-11-01097-f001:**
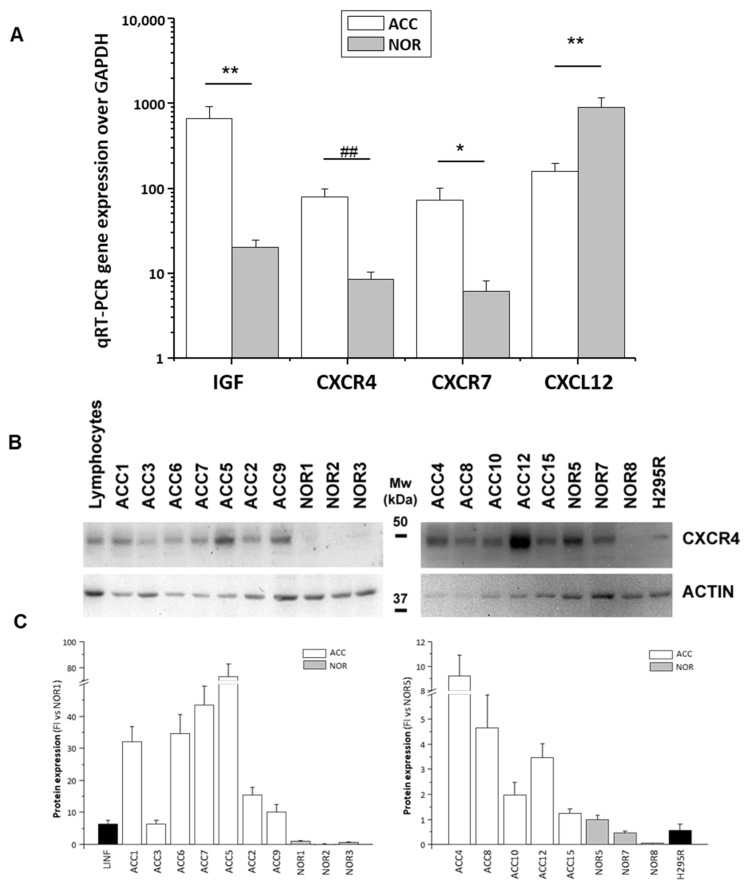
Differential expression of IGF-II and CXCR4/CXCR7/CXCL12 in ACC and normal adrenals. (**A**) Gene expression measured by qRT-PCR Taqman assay in ACC biopsies from n = 22 patients (ACC) and in normal adrenal samples (NOR, n = 26). Data are expressed as mean±SE gene expression normalized on the house-keeping gene GAPDH. Statistical significance between the two groups of data after Student’s *t* test analysis: * *p* < 0.05, ** *p* < 0.01, and ## *p* < 0.001 ACC vs. NOR. (**B**) Representative Western blot analysis of CXCR4 protein in lysates of ACC and normal adrenal biopsies from different subjects, as well as in H295R cell and human lymphocyte lysates, used as positive control. The membranes were re-probed for actin for loading control. Molecular weight markers (Mw) are indicated. (**C**) Densitometric analysis of protein expression as detected by Western blot technique: data are expressed as mean ± SE fold increase (FI) versus NOR of CXCR4 band intensity normalized on actin from n = 3 independent experiments.

**Figure 2 jpm-11-01097-f002:**
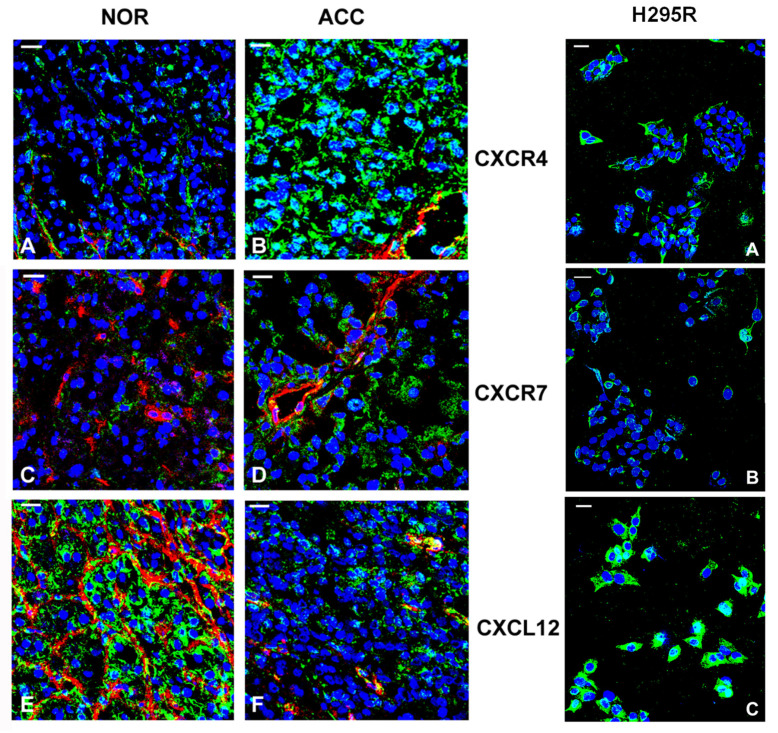
CXCR4/CXCR7 and CXCL12 expression and localization in ACC, normal adrenal cortex and H295R cell line. Laser confocal microscopy analysis. Left panel. Representative images of normal adrenal cortex (NOR) or ACC biopsies showing positivity (green fluorescence) of CXCR4 (**A**,**B**), CXCR7 (**C**,**D**), and CXCL12 (**E**,**F**) in adrenal malignant and normal cells and in endothelial cells of the vasculature structures, the latter also co-stained with anti-vWf (**A**,**B**,**E**,**F**) or anti-CD31 (**C**,**D**) antibodies (red fluorescence). Yellow fluorescence indicates co-localization between CXCR4, CXCR7, or CXCL12 expression (green fluorescence) and vWF or CD31 (red fluorescence) in endothelial cells. Nuclei were counterstained with Topro-3 (blue). Confocal analysis was performed in the biopsies of five normal adrenals and ACCs of five patients. Right panel. Representative images from three independent H295R cell cultures showing the staining (green) for CXCR4 (**A**), CXCR7 (**B**) and CXCL12 (**C**). Nuclei were counterstained with Topro-3 (blue). Bar 20 µm. The receptor staining seems to be discrete and mainly localized at membrane level, whereas CXCL12 positivity is more diffuse at the cell surface.

**Figure 3 jpm-11-01097-f003:**
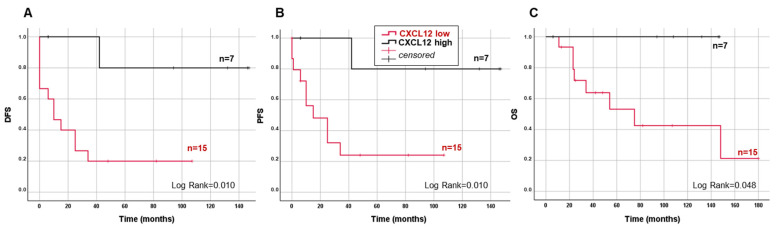
Survival predictive value of CXCL12 gene expression in ACC. Kaplan–Meier analysis of DFS (**A**), PFS (**B,**) and OS (**C**) in ACC patients stratified into two classes according to low- and high-CXCL12 gene expression (cut off = 169 2^−ΔΔCt^, according to upper tertile of CXLC12 mRNA level distribution in the n = 22 ACCs analyzed). Statistical significance is indicated by log rank.

**Figure 4 jpm-11-01097-f004:**
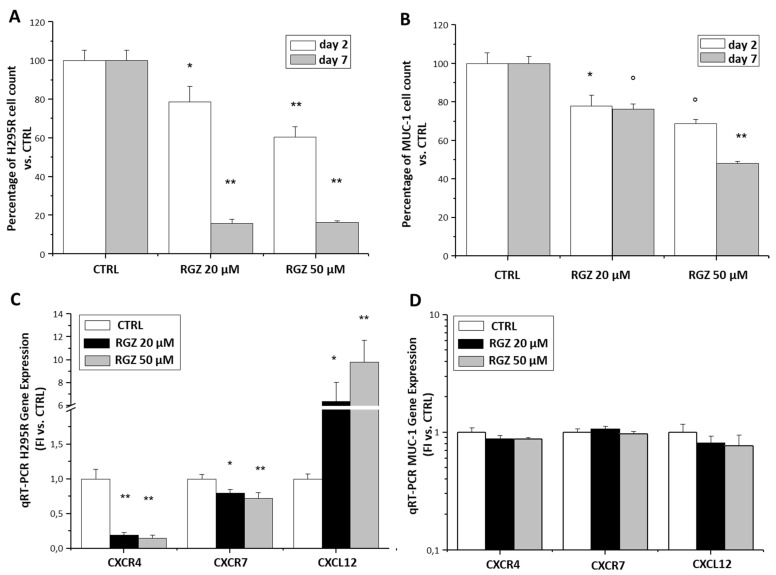
RGZ effects on proliferation and gene expression of CXCL12/CXCR4 axis in adrenocortical cell lines. Different doses (20 and 50 µM) of RGZ time-dependently inhibited H295R (**A**) and MUC-1 (**B**) cell growth. Data are expressed as the mean percentage of cell growth ± SE versus the respective Ctrl taken as 100%, at two different time points of treatment (2 and 7 days). Statistical significance obtained by ANOVA followed by Dunnett’s post-hoc test: * *p* < 0.05, ° *p* < 0.01 and ** *p* < 0.001 vs. respective Ctrl. (**C**,**D**) RGZ inhibited CXCR4 and 7- and stimulated CXCL12 expression in H295R cells (**C**) but not in MUC-1 (**D**) as detected by qRT-PCR TaqMan analysis of CXCR4, CXCR7, and CXCL12 gene expression in cells treated for 24 h with RGZ at different doses (20 and 50 µM). Data are expressed as the mean fold increase in gene expression normalized on GAPDH ± SE. Statistical significance obtained by ANOVA followed by Dunnett’s post-hoc test: * *p* < 0.05 and ** *p* < 0.001 vs. respective Ctrl.

**Figure 5 jpm-11-01097-f005:**
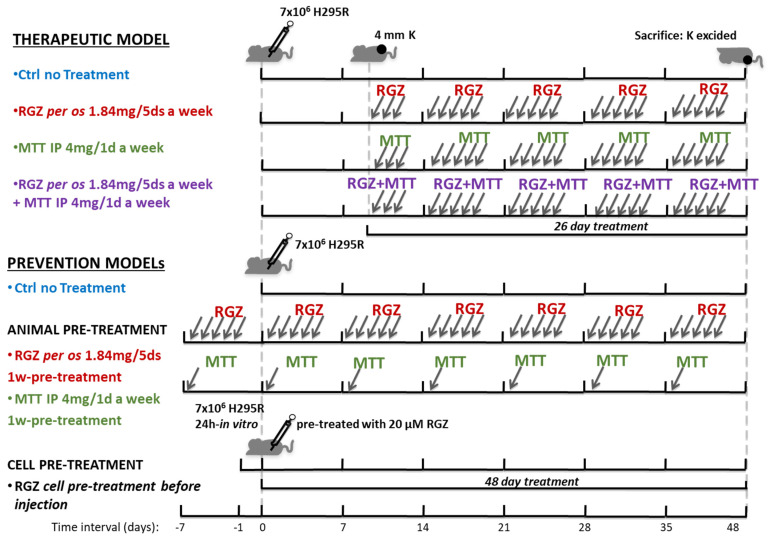
Scheme of mouse ACC xenograft therapeutic and prevention models for RGZ and MTT treatment. ACC therapeutic models of xenografts were obtained by H295R cell subcutaneous injection (7.5 × 10^6^ cells) in CD1 nude mice. Six animals in each group of treatment, including the untreated controls, were used. Once tumors had reached a detectable 4 mm diameter, the animals were randomized to be left untreated (Ctrl) or treated with RGZ (1.84 mg per os 5 days a week) or with MTT (4 mg intraperitoneal 1 day a week) or a combination of the two drugs. After 26 days of treatment, the animals were sacrificed. Two different prevention models of xenograft were developed. The animal pre-treatment model was obtained by one-week pre-treatment of mice with RGZ (1.84 mg per os for five days) or with MTT (4 mg intraperitoneal for one day), before H295R cell injection (7.5 × 10^6^ cells). RGZ and MTT treatment was continued for an additional 48 days, and then mice were sacrificed. A second prevention model (cell pre-treatment model) was obtained by 24 h-H295R cell in vitro pre-treatment with 20 µM RGZ before cells were subcutaneously injected in CD1 nude mice (7.5 × 10^6^ cells). Another group of mice was injected with the same number of non-pre-treated H295R cells (7.5 × 10^6^ cells) and used as Ctrls for both prevention models (control). K indicates the tumor mass; IP = intraperitoneal, per os = per os gavage.

**Figure 6 jpm-11-01097-f006:**
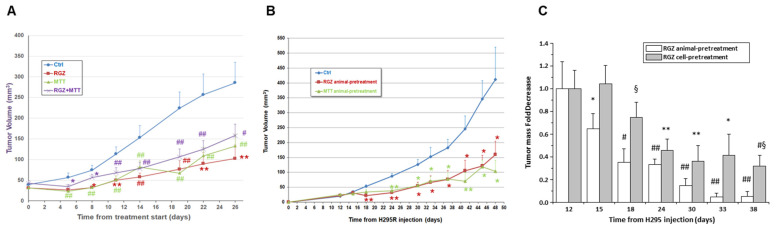
RGZ and MTT inhibit tumor growth in therapeutic and prevention mouse xenograft models of ACC. (**A**) In ACC therapeutic models, tumor growth was followed for an additional 26 days from cell subcutaneous injection by measuring the mass volume 3 times a week. Tumor growth curves represent mean volume ± SE over time. Statistical difference between each treatment and the control curve was assessed at any time point by Student’s *t* test (* *p* <0.05, ** *p* < 0.01, # *p* < 0.005, ## *p* < 0.001). No statistically significant difference among treatments was observed at any time point by ANOVA analysis of variance. (**B**) In the ACC pre-treatment animal model, tumor growth was assessed by measuring the mass volume at the site of injection 3 times a week for an additional 48 days from the day of cell subcutaneous injection. Tumor-growth curves represent mean volume ± SE over time. Statistical difference between each treatment and the control curve (untreated animals injected with H295R cells on the same day as the pre-treated group) were assessed at any time point by Student’s *t* test (* *p* < 0.05 and ** *p* < 0.01). No statistically significant difference between the two types of treatment was observed at any time point by Student’s *t* test. (**C**) In the cell pre-treatment model, tumor growth was followed for an additional 38 days from the day of cell subcutaneous injection. Data are expressed as media ± SE of the tumor-mass-fold decrease in the two pre-treatment models versus tumor mass in control curve (untreated animals injected with H295R cells on the same day as the pre-treated groups) at any time point. Tumor mass in all groups was initially expressed as fold increase over the tumor mass at 12 days from cell injection calculated in each mouse. Statistical difference between tumor mass in the pre-treatment models and the control was assessed at any time point by Student’s *t* test (* *p* < 0.05, ** *p* < 0.01, # *p* < 0.005, ## *p* < 0.001 vs. untreated Ctrl; § *p* < 0.05 animal- vs. cell-pre-treatment). See also Figure 5.

**Figure 7 jpm-11-01097-f007:**
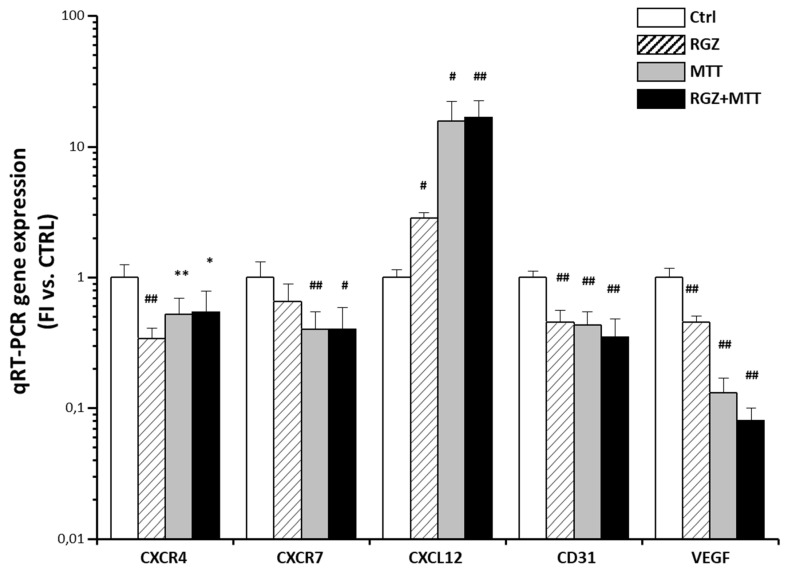
RGZ and MTT effects on XCL12/CXCR4 axis and angiogenesis in the tumor mass in the xenograft therapeutic model. The qRT-PCR Taqman analysis of human CXCR4, CXCR7, CXCL12 and murine CD31, VEGF gene expression in the subcutaneous tumors excided from the xenograft mice in the therapeutic model. Explants from at least four independent mice in each group of treatment, including the untreated controls were extracted for qRT-PCR analysis. Data are expressed as mean fold increase in gene expression ± SD normalized on GAPDH or RSP18, for expression of human (CXCR4, CXCR7, CXCL12) or murine genes (CD31, VEGF), respectively. Statistical analysis was performed by Student’s *t* test between each class of treatment and the untreated Ctrl (* *p* < 0.05, ** *p* < 0.01, # *p* < 0.005, ## *p* < 0.001).

**Table 1 jpm-11-01097-t001:** Anthropometrical and clinical parameters of ACC patients (n = 22). Data are reported for the total cohort and for patients stratified by levels of CXCL12 expression (low and high CXCL12). Only statistically significant P values are reported after Student’s test or χ^2^ test for continuous or noncontinuous variables, as well as after U-Mann–Whitney test for nonparametric variables. Patients’ follow-up interval was between 0 and 180 months from surgery. IQR: interquartile range.

Patients Characteristics	Total Cohort (n = 22)	Low CXCL12 (n = 15)	High CXCL12 (n = 7)	*p*
Sex: female, n (%)	17 (77)	11 (73)	6 (86)	ns
Age at diagnosis (ys), mean ± SE	48.9 ± 3.3	53.1 ± 3.8	38.3 ± 4.6	0.041
ENSAT stage, n (%)				0.05
I	3 (14)	1 (7)	2 (29)	
II	5 (23)	2 (13)	3 (43)	
III	9 (41)	8 (53)	1 (14)	
IV	4 (18)	4 (27)	0 (0)	
Unknown	1 (4)	0 (0)	1 (14)	
Metastasis/Recurrence, n (%)	13 (59)	12 (80)	1 (14)	0.004
no information	0 (0)	0 (0)	0 (0)	
Death for ACC, n (%)	9 (41)	9 (60)	0 (0)	ns
no information	0 (0)	0 (0)	0 (0)	
DFS time (months), median [IQR]	20 (4.5–85)	10 (0–34)	94 (6–146)	0.039
PFS time (months), median [IQR]	20 (6–85)	10 (1–34)	94 (6–146)	0.047
OS time (months), median [IQR]	51 (23–114)	42 (23–82)	108 (6–146)	ns
Tumor size (cm), mean ± SE	9.7 ± 1.0	11.0 ± 1.1	6.3 ± 1.5	0.035
Hormone secretion, n (%)				ns
Yes	12 (54)	9 (60)	3 (43)	
Cortisol	5 (23)	5 (33)	0 (0)	
Sex steroids	7 (32)	4 (27)	3 (32)	
Mineralcorticoids	1 (0)	1 (7)	0 (0)	
No	7 (32)	5 (40)	1 (15)	
Unknown	3 (14)	0 (0)	3 (43)	
Ki67 LI (%), mean ± SE	18.1 ± 4.8	22.3 ± 6.5	10.4 ± 5.5	ns
Unknown, n (%)	2 (9)	2 (13)	0 (0)	
Weiss score, mean ± SE	5.8 ± 1.8	6.7 ± 0.4	4.3 ± 0.6	0.002
Unknown, n (%)	2 (9)	2 (13)	0 (0)	
Resection status, n (%)				ns
R0	11 (50)	8 (54)	3 (43)	
R > 0	5 (23)	5 (33)	0 (0)	
Unknown	6 (27)	2 (13)	4 (57)	
Mitotane, n (%)				0.005
Yes	14 (64)	13 (87)	1 (14)	
No	6 (27)	2 (13)	4 (57)	
Unknown	2 (9)	0 (0)	2 (29)	
Other Chemotherapy (EDP), n (%)				ns
Yes	5 (22)	5 (33)	0 (0)	
No	15 (68)	9 (60)	6 (86)	
Unknown	2 (9)	1 (7)	1 (14)	
Radiotherapy, n (%)				ns
Yes	2 (9)	2 (9)	0 (9)	
No	18 (91)	13 (91)	7 (91)	
Unknown	2 (0)	0 (0)	0 (0)	

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
