# Peer review of "Stimulated Expression of CXCL12 in Adrenocortical Carcinoma by the PPARgamma Ligand Rosiglitazone Impairs Cancer Progression"

_jpm, 2021, doi:10.3390/jpm11111097_

Round 1

Reviewer 1 Report

Cantini et al. report on the investigation that ACC do not only demonstrate CXCR4 and CXCR7, but also expression of the respective ligand 21 (CXCL12). Clinically, they show that CXCL12 is negatively correlated with DFS, PFS and OS. Moreover, Rosiglitazone treatment reduced H295R cell number and CXCR4 and 7 receptor expression together with accompagnied upregulation of CXCL12. Three different settings of H295R-xenografts were tested (therapeutic, pre-treatment animals, pre-treatment cells) in vivo for which tumor growth inhibition was observed upon rosiglitazone and mitotane treamtments. No additive effects of a combination treatment were detected.  H295R-tumor growth inhibition was associated with suppression of CXCR4/CXCR7 and stimulation of human CXCL12 expression.

In general the study is of interest. However, while most of the CXCR4 and 7 related data have been already published previously by Chifu ez al. 2020, for the reported novel findings on CXCL12 and therapeutic regulations the current manuscript lacks in depth investigation in patient samples and/or regarding appropriate further tumor models. The authors declare that underlying regulations might be involved in tumor progression and metastasis, but even though further models including some of metastatic origin are nowadays available, all preclinical experiments are performed basing on H295R only, which is derived from a primary tumor.

Specific comments:

- For all types of experiments H295R was implemented exclusively, while in this specific context it could be of great interest to compare with models of metastatic origin. In 2016 Hantel et al reported in Oncotarget on the development of another tumor model providing both, cell line and tissue-based Xenografts for a model derived from an ACC neck metastasis. In 2018 Kiseljak-Vasiliades et al. published cell lines and xenografts named CU-ACC 1 and 2 in Endocr. Rel. Cancer, in 2021 Landwehr et al. the cell line JIL-2266 in Eur J Endocrinol. Thus, nowadays there are additional models available including some of metastatic origin. Up to my knowledge all mentioned research groups provide for collaborative studies their material to other researchers. Thus, in my opinion it would be of great benefit to include a second ACC model at least for some additional in vitro experiments to provide a comparison vs. the primary tumor derived NCI-H295 cells.

-Fig 6: If possible, the same graphs should be provided for all studies (either tumor fold or tumor volume) incl also those for both pre-treatments. If not possible, I would kindly ask the authors to comment on the reasons. In general, I would prefer the tumor volume as tumor mass folds can sometimes lead to the impression of huge therapeutic effects which are basing on very small changes in the tumor development. Other figures/analyses can be nicely given as additional analysis. However, if possible I would like to have a look on the tumor growth curves (Volume) first.

-Fig 7: why are students tests and not One Way Anovas incl Bonferroni/Dunnett post tests provided for the respective groups of the four treatments? Moreover, i am surprised that some values completely lack error bars, please check that this is indeed the case and did not accidentally get lost via ongoing analysis (clicking on all bars instead of one or reversly – what quickly happens).

Minor comments:

-Overall clinical parameters are given in table 1, DFS, PFS and OS correlated in figure 3. Other correlation outcomes are given in  the text, but I can not see an appropriate data table for that. Maybe an additional high/low comparison table could be added or table 1 extended in this regard.

-Correct to 7 x 106 cells instead of 7 x 106 (e.g in fig 5)

- I would propose to provide graphs with SEMs instead of SDs. Some statistical analysis would be better comprehensible such as in fig 1A (CXCR4 vs7).

- An additional quantification of the WB in Fig 1B would be of benefit

- In figure 6A the final curve descent in all three treatment groups appears unusual to me. From my experience if xenografts showed over a longer time already a continuous tumor increase under therapeutic treatments (as seen up to day 26 in all groups) it is rare that this changes after more than 4 weekly cycles of continuous drug administration and then simultaneously in three different treatment groups at the same day. Thus, I would rather think of something methodologically here, comparable effects are seen for tumor volume in fig 6B from day 42 to day 44 and back to 46. However, depending on the tumor phenotype it is just hard to always measure exatly the same dimensions of the xenografts. However, this only as separate comment. As the authors do not speculate on significant differences between the treatment groups and as the single groups reached already statistical significance compared to untreated controls there is no strong effect on the discussed outcome.

Author Response

POINT BY POINT ANSWER TO REVIEWERS
We thanks the Reviewers and Editors for their careful reading of the manuscript and for the constructive comments raised. Accordingly, we have extensively edited the manuscript and made the required changes. All the changes made in the Text, Figures and Table have been highlighted. In particular, we performed the experiments on the second cell line MUC-1 and added the results as new panels in Fig. 4. MUC-1 cells have
been kindly provided by Dr. Constanze Hantel (University of Zurich). Please note, that she has been added in the revised version of the manuscript as a co-author.

We hope that the present version of the manuscript could be found publishable in the Journal.

REVIEWER 1

Comments and Suggestions for Authors

Cantini et al. report on the investigation that ACC do not only demonstrate CXCR4 and CXCR7, but also expression of the respective ligand 21 (CXCL12). Clinically, they show that CXCL12 is negatively correlated with DFS, PFS and OS. Moreover, Rosiglitazone treatment reduced H295R cell number and CXCR4 and 7 receptor expression together with accompanied upregulation of CXCL12. Three different settings of H295R-xenografts were tested (therapeutic, pre-treatment animals, pre-treatment cells) in vivo for which tumor growth inhibition was observed upon rosiglitazone and mitotane treatments. No additive effects of a combination treatment were detected. H295R-tumor growth inhibition was associated with suppression of CXCR4/CXCR7 and stimulation of human CXCL12 expression.

In general, the study is of interest. However, while most of the CXCR4 and 7 related data have been already published previously by Chifu et al. 2020, for the reported novel findings on CXCL12 and therapeutic regulations the current manuscript lacks in depth investigation in patient samples and/or regarding
appropriate further tumor models. The authors declare that underlying regulations might be involved in tumor progression and metastasis, but even though further models including some of metastatic origin are nowadays available, all preclinical experiments are performed basing on H295R only, which is derived from a primary tumor.

Compared to Chifu et al 2020, which was focused on the analysis of the CXCR4/CXCR7 cognate receptors in ACC, our study demonstrates for the first time the ability of the primary tumor cells to produce the ligand CXCL12, which might be involved in contributing to the modulation of the metastatic potential of these cells, as suggested by the correlation of CXCL12 tumor expression with PFS and OS in the cohort of ACC patients analyzed. Moreover, we also demonstrate that RGZ treatment results in upregulating CXCL12 expression and
downregulating CXCR4/7 expression in the primary tumor cell lines.

Finally, according to the Reviewer’s suggestion, we also extended the analysis of CXCL12 and its cognate receptor expression in basal conditions and under RGZ stimulation, to the metastatic ACC cell line, MUC-1.

Specific comments:

- For all types of experiments H295R was implemented exclusively, while in this specific context it could be of great interest to compare with models of metastatic origin. In 2016 Hantel et al reported in Oncotarget on the development of another tumor model providing both, cell line and tissue-based Xenografts for a model derived from an ACC neck metastasis. In 2018 Kiseljak-Vasiliades et al. published cell lines and xenografts named CU-ACC 1 and 2 in Endocr. Rel. Cancer, in 2021 Landwehr et al. the cell line JIL-2266 in Eur J Endocrinol. Thus,
nowadays there are additional models available including some of metastatic origin. Up to my knowledge all mentioned research groups provide for collaborative studies their material to other researchers. Thus, in my opinion it would be of great benefit to include a second ACC model at least for some additional in vitro experiments to provide a comparison vs. the primary tumor derived NCI-H295 cells.

We thank the Reviewer for their suggestion to also perform the in vitro experiments in another ACC cell line, MUC-1, to compare the behaviour of primary tumor and metastatic cell lines.

MUC-1 have been kindly provided by Dr. C. Hantel (University of Zurich), who is now co-author of the paper.

We have now analyzed CXCR4/7 and CXCL12 expression in MUC-1 in basal conditions and in response to RGZ
treatment in comparison with H295R, also assessing RGZ effects on cell growth in vitro.

RGZ dose-dependently inhibited in vitro proliferation of the adrenocortical carcinoma primary tumor cell line,
H295R, as well as of the metastatic cell line, MUC-1. The effect was similar between the two cell lines at two days, while MUC-1 seemed to be more resistant than H295R in the longer period. A different sensitivity in the response to cytotoxic agents has already been reported for MUC-1 and H295R, with MUC-1 showing to be more chemoresistant than the primary tumor cells, probably due to their metastatic origin. The reduced
proliferation was paralleled by a significant increase in CXCL12 gene expression and decrease of its two receptors only in H295R. Conversely, in MUC-1, RGZ did not modulate the CXCL12/CXCR4 axis, suggesting that the up-regulated CXCL12 in response to RGZ might be active only in the primary tumor but not in metastatic cells. These findings together with the high levels of CXCL12 in the tumor associated with a lower risk of
metastasis and progression, suggest that in the primary tumor CXCL12 endogenously produced by the tumor cells might act in a paracrine manner to contain progression, making tumor cells less sensitive to the CXCL12 gradient that drives the metastatic process. Notably, despite a similar basal expression of CXCL12 between MUC-1 and H295R, the primary tumor cells display higher levels of CXCR7 and the metastatic cells of CXCR4.

These new data have now been included in new Fig. 4.

-Fig 6: If possible, the same graphs should be provided for all studies (either tumor fold or tumor volume) incl also those for both pre-treatments. If not possible, I would kindly ask the authors to comment on the reasons.
In general, I would prefer the tumor volume as tumor mass folds can sometimes lead to the impression of huge therapeutic effects which are basing on very small changes in the tumor development. Other figures/analyses can be nicely given as additional analysis. However, if possible I would like to have a look on the tumor growth curves (Volume) first.

Since there was a variability in the initial dimension of the tumor mass among the different sets of experiments that were performed, we thought the best way to compare the data was to express them as tumor volume fold increase over the time 0 of treatment for each animal. This is particularly evident for the graph in Fig.6 panel C where the inhibitory effects of the two different pretreatment sets (RGZ cell pretreatment and RGZ
animal pretreatment) were compared. This is why in the revised version, we maintained this type of expression of the tumor growth (tumor growth fold increase over time 0 for each animal) for panel C curves. Instead, we uniformed the two tumor growth panels (A and B), reporting the mean of tumor volume between all animals in the same group at any time point. In panel A, we also delated the last time point of treatment (29 days), as requested by this Reviewer in another comment (see below) and recalculated the percentage of growth inhibition for all treatments in the therapeutic setting.

-Fig 7: why are students tests and not One Way Anovas incl Bonferroni/Dunnett post tests provided for the respective groups of the four treatments?

In agreement with the statistical analysis applied for tumor growth in the 4 classes of xenograft treatments (Fig. 6A), we decided to compared each treatment class with the control one, as we are not interested in evaluating also the difference between the other treatment classes as the final effect on growth was similar among them.

Moreover, I am surprised that some values completely lack error bars, please check that this is indeed the case and did not accidentally get lost via ongoing analysis (clicking on all bars instead of one or reversely what quickly happens).

We cannot see this point, as in Fig.7 all bars had SD bars. Please note, that in agreement with this Reviewer’s comment all SD error bars have been replaced by SE. Please note that in some cases, SE is such small that it seems not visible.

Minor comments:

-Overall clinical parameters are given in table 1, DFS, PFS and OS correlated in figure 3. Other correlation outcomes are given in the text, but I can not see an appropriate data table for that. Maybe an additional high/low comparison table could be added or table 1 extended in this regard.

The suggested table has now been added as new Table 1, reporting the main parameters distribution in the two classes (low and high CXCL12 expression) and the statistical significance.

-Correct to 7 x 106 cells instead of 7 x 106 (e.g in fig 5)

Done

- I would propose to provide graphs with SEMs instead of SDs. Some statistical analysis would be better comprehensible such as in fig 1A (CXCR4 vs7).

Done

- An additional quantification of the WB in Fig 1B would be of benefit.

Quantification of the bands in different WBs has now been performed and the mean fold increase values over normal adrenal samples (NOR) added as graph in Fig.1C. Moreover, fold increase expression of CXCR4 in ACC vs Normal adrenal samples and its statistical significance based on WB quantification has now been provided in the text.

- In figure 6A the final curve descent in all three treatment groups appears unusual to me. From my experience if xenografts showed over a longer time already a continuous tumor increase under therapeutic treatments (as seen up to day 26 in all groups) it is rare that this changes after more than 4 weekly cycles of continuous drug administration and then simultaneously in three different treatment groups at the same day. Thus, I would rather think of something methodologically here, comparable effects are seen for tumor volume in fig
6B from day 42 to day 44 and back to 46. However, depending on the tumor phenotype it is just hard to always measure exactly the same dimensions of the xenografts. However, this only as separate comment. As the authors do not speculate on significant differences between the treatment groups and as the single groups reached already statistical significance compared to untreated controls there is no strong effect on the discussed outcome.

In line with this Reviewer’s comment and since this simultaneous drop in the tumor growth in all the 3 classes
of treatment is difficult to be interpreted, we decided to delate the data corresponding to 29 day-treatments in the 4 classes in Fig. 6A.

Reviewer 2 Report

This is an interesting work with potential important therapeutic implications.

I only have some minor issues to place:

Line 21: The sentence is a bit tricky. Could you please rephrase?

Line 131: PFS log-rank test is not indicated. 

Lines 157-158: "... reduced the cell number with increasing effect with longer ...". Not so clear, please rephrase.

Lines 131-147: Median follow-up time should be indicated. Furthermore, add median values of survival outcomes in months.

Lines 349-350: Are ACC tissues taken at initial diagnosis or later time-points?

Discussion:

The story of CXCR4 in cancer starts with the seminal work of Muller et al published in 2001 on Nature. Since then, however, chemokines have not produced striking advancements either in patient stratification or treatment.

I would suggest to highlight two comments:

1 - CXCL12(SDF-1) gene expression is a target of direct or paracrine 17b-estradiol action in breast cancer. Could estradiol have any effect on adrenal cortex?

2 -   Plerixafor is another selective reversible antagonist of CXCR4 and it is commonly used in bone marrow transplantation for stem cells mobilization in the bloodstream. Could it be tested in ACC?

Materials and Methods + Results:

As many researchers (like me) experienced problems with CXCR4 MoAbs in immuno-histochemical staining, I would ask to add the dilution that was used for anti-CXCR4, anti-CXCR7 and anti-CXCL12 and if that used was the dilution indicated by the manifacturer of different dilutions were tested. In the results section I would also appreciate a description of the type of staining within the ACC tissue: ie membrane, citoplasm, nucleus staining? All of them? Diffuse or granular?

Author Response

POINT BY POINT ANSWER TO REVIEWERS
We thanks the Reviewers and Editors for their careful reading of the manuscript and for the constructive comments raised. Accordingly, we have extensively edited the manuscript and made the required changes. All the changes made in the Text, Figures and Table have been highlighted. In particular, we performed the experiments on the second cell line MUC-1 and added the results as new panels in Fig. 4. MUC-1 cells have
been kindly provided by Dr. Constanze Hantel (University of Zurich). Please note, that she has been added in the revised version of the manuscript as a co-author.

We hope that the present version of the manuscript could be found publishable in the Journal.

REVIEWER 2

Comments and Suggestions for Authors

This is an interesting work with potential important therapeutic implications.

I only have some minor issues to place:

Line 21: The sentence is a bit tricky. Could you please rephrase?

Done

Line 131: PFS log-rank test is not indicated.
Done

Lines 157-158: "... reduced the cell number with increasing effect with longer ...". Not so clear, please rephrase.

Done

Lines 131-147: Median follow-up time should be indicated. Furthermore, add median values of survival outcomes in months.

Median time of OS, PFS and DFS with interquartile range (IQR) are now indicated in Table 1 for the entire cohort as well as for the two subgroups stratified according to CXCL12 levels, with statistics performed applying U-Mann Whitney test for nonparametric variables. The follow-up interval for the entire cohort was 0-180 months, as now added in Table 1 legend. Moreover, the main parameters distribution in the two classes
(low and high CXCL12) have been added to Table 1.

Lines 349-350: Are ACC tissues taken at initial diagnosis or later time-points?

All tumor samples have been taken during tumor removal, which corresponds to the time point ACC diagnosis
is made by the pathologist.

Discussion:

The story of CXCR4 in cancer starts with the seminal work of Muller et al published in 2001 on Nature. Since then, however, chemokines have not produced striking advancements either in patient stratification or treatment.

This may depend on the type of cancers. Here, we demonstrate that CXCL12 may represent a valuable additional parameter to refine stratification of patients in ACC.

I would suggest to highlight two comments:

1 - CXCL12(SDF-1) gene expression is a target of direct or paracrine 17b-estradiol action in breast cancer. Could estradiol have any effect on adrenal cortex?

This is an important note: indeed, estrogens have been demonstrated to induce CXCL12 production in endometrial and breast cancer cells through activation of ERalpha (Hall et al, Mol Endocrinol. 2003). The effect of estrogens is still debated in ACC and currently under study. Indeed, the involvement of both classical ERs as well as estrogen-related receptors (ERR) and GPER has been demonstrated to be expressed in ACC tumors and involved in regulation of ACC growth and progression (Barzon et al, Virchows Arch. 2008; Sirianni et al, J Clin
Endocrinol Metab. 2012; Felizola et al, Mol Cell Endocrinol. 2013; Chimento et al, Oncotarget. 2015; Casaburi et al, Front Endocrinol 2018). Interestingly, mitotane effects have been suggested to be partially mediated by estrogen receptors in ovary and ACC (Nader et al JCEM 2006; Rossini et al, Biomedicine 2021), thus we could speculate that the increased levels of CXCL12 observed in response to mitotane in the xenograft models might involve activation of the estrogenic pathway. A comment has been added in the discussion.

2 - Plerixafor is another selective reversible antagonist of CXCR4 and it is commonly used in bone marrow transplantation for stem cells mobilization in the bloodstream. Could it be tested in ACC?

Plerixaflor and other CXCR4 antagonists are mainly used to mobilize hematopoietic stem cells in cell apheresis in multiple myeloma and lymphoma (De Clercq E. Antivir Chem Chemother 2019), and might be tested in ACC once this anti-cancer strategy is recognized to be applicable to ACC. However, their use is still under evaluation and specific trials are required.

Materials and Methods + Results:

As many researchers (like me) experienced problems with CXCR4 MoAbs in immuno-histochemical staining, I would ask to add the dilution that was used for anti-CXCR4, anti-CXCR7 and anti-CXCL12 and if that used was the dilution indicated by the manifacturer of different dilutions were tested.

The antibody work dilution we used for immunohistochemistry has now been indicated for the 3 antibodies in Methods section as requested.

In the results section I would also appreciate a description of the type of staining within the ACC tissue: ie membrane, citoplasm, nucleus staining? All of them? Diffuse or granular?

We have now added a comment on the staining localization and type of staining as detectable in immuno-histochemistry and -cytochemistry in Fig. 2 legend. The receptor staining seems to be discrete and mainly localized at membrane level, whereas CXCL12 is more diffuse at the cell surface.
